# Nationwide Screening for Bee Viruses in *Apis mellifera* Colonies in Egypt

**DOI:** 10.3390/insects14020172

**Published:** 2023-02-09

**Authors:** Mohamed Kandel, Robert J. Paxton, Yahya Al Naggar

**Affiliations:** 1Plant Protection and Molecular Diagnosis, Arid Lands Cultivation Research Institute, City of Scientific Research and Technological Applications (SRTA-City), Alexandria 21934, Egypt; 2General Zoology, Institute for Biology, Martin Luther University Halle-Wittenberg, Hoher Weg 8, 06120 Halle (Saale), Germany; 3Zoology Department, Faculty of Science, Tanta University, Tanta 31527, Egypt

**Keywords:** honey bee viruses, Egypt, deformed wing virus, prevalence, varroa mite

## Abstract

**Simple Summary:**

Diseases, particularly those caused by viruses, are a major cause of bee colony losses. However, little is known about the prevalence of the pathogens, particularly virus prevalence, of the honey bee in Egypt, one of the most important countries for beekeeping and agricultural production in Africa. To address this shortfall, we determined the prevalence of ten widespread bee viruses in honey bee colonies in Egypt between two major season, winter and summer, in relation to colony infestation by varroa mites.

**Abstract:**

Honey bees are essential for crop and wild plant pollination. However, many countries have reported high annual colony losses caused by multiple possible stressors. Diseases, particularly those caused by viruses, are a major cause of colony losses. However, little is known about the prevalence of honey bee pathogens, particularly virus prevalence, in Egyptian honey bees. To address this shortfall, we determined the prevalence of widespread bee viruses in honey bee colonies in Egypt—whether it is affected by geography, the season, or infestation with *Varroa destructor* (varroa) mites. Honey bee worker samples were collected from 18 geographical regions across Egypt during two seasons: winter and summer of 2021. Three apiaries were chosen in each region, and a pooled sample of 150 worker bees was collected from five colonies in each apiary then screened by qPCR for 10 viral targets: acute bee paralysis virus (ABPV), black queen cell virus (BQCV), chronic bee paralysis virus (CBPV), deformed wing virus (DWV) genotypes A (DWV-A), B (DWV-B) and D (Egyptian bee virus), Israeli acute paralysis virus (IAPV), Kashmir bee virus (KBV), sacbrood virus (SBV), and slow bee paralysis virus (SBPV). Our results revealed that DWV-A was the most prevalent virus, followed by BQCV and ABPV; the DWV genotype now spreading across the world, DWV-B, was not detected. There was no difference in varroa infestation rates as well as virus prevalence between winter and summer. However, colonies infected with BQCV had a significantly higher varroa count (adjusted *p* < 0.05) in the winter season, indicating that there is a seasonal association between the intensity of infestation by varroa and the presence of this virus. We provide data on the current virus prevalence in Egypt, which could assist in the protection of Egypt’s beekeeping industry. Moreover, our study aids in the systematic assessment of the global honey bee virome by filling a knowledge gap about the prevalence of honey bee viruses in Egypt.

## 1. Introduction

Honey bees play an important role in crop and wild plant pollination [1,2]. However, since 2006, they have suffered significant overwinter colony losses, particularly in Europe and the United States [3,4,5]. These declines have been linked to parasites, pathogens, poor nutrition, and pesticide exposure [6,7,8]. Among these pathogens are viruses, including several that are transmitted by exotic varroa (*Varroa destructor*) mites, which are themselves widely recognised as a major cause of colony death and which can act synergistically with other biotic and abiotic stressors, leading to the collapse of host colonies [9]. Additionally, there is growing evidence that viruses infecting honey bees are found in a wide range of other pollinators species and therefore threaten our ecosystems more profoundly than previously thought [10].

Over twenty-four viruses have been identified in honey bees [11], most of which are in the order Picornavirales [12,13]. These include the common bee viruses which have positive single-strand RNA genomes (+ssRNA), either in the family Dicistroviridae, which includes acute bee paralysis virus (ABPV), black queen cell virus (BQCV), Israeli acute paralysis virus (IAPV), and Kashmir bee virus (KBV), or in the family Iflaviridae, which includes deformed wing virus (DWV) genotypes A (DWV-A) and B (DWV-B) (the latter is otherwise known as *Varroa destructor* virus-1), slow bee paralysis virus (SBPV), and sacbrood virus (SBV). Several of these can be transmitted by varroa mites, including ABPV, DWV-A, DWV-B, IAPV, KBV, and SBPV [14]. Additional characterized +ssRNA viruses include Lake Sinai viruses (LSV), which are in the Sinai virus genus [15]. It has been reported that the prevalence of honey bee viruses and other pathogens varies across habitats in relation to their nutritional resources [16] and is also influenced by climate, leading to patterns of prevalence that vary between seasons [17]. 

According to the FAO database, Egypt experienced a 50% decrease in the number of honey bee colonies between 2005 and 2016 [18], a decline that coincided with colony losses observed in the USA and Europe [16]. This is clear evidence of the immediate current threat to pollination in Egypt. This rate of colony losses is expected to worsen in the future if it is not counteracted [18]. It is therefore important to determine the causes of Egypt’s honey bee decline. Many countries have reported high annual colony losses caused by multiple stressors [19], including DWV, now widespread across the world [20]. Although numerous studies have been conducted to assess the prevalence of viruses in honey bee colonies around the world [9], there has only been one study to date that has screened some honey bee viruses in Egyptian bee colonies [21], and which reported that BQCV, DWV-A, and IAPV were widespread, whilst DWV-B was present in Egyptian honey bees, though without specifying the geographic distribution of these viruses. As a result, more comprehensive studies involving different geographical regions and seasons are required to determine the prevalence of bee viruses in Egyptian bee colonies in order to understand and interpret observed colony losses.

The purpose of this study was to investigate the prevalence of ten common bee viruses in honey bee colonies across Egypt and between two major season, winter and summer, in relation to colony infestation by varroa mites. 

## 2. Materials and Methods

### 2.1. Honey Bee Samples

Honey bees were collected from 54 apiaries in 18 governorates at 2 time points—during the winter (January–February) and during the summer (June–August) of 2021. A selection of 3 apiaries separated by at least 10 km were chosen in each governorate (Figure 1), with a composite pooled sample of approximately 150 worker bees equally derived from 5 colonies (ca. 30 worker bees per colony) per apiary. Honey bee workers were collected from both the outer and brood frames and at the hive entrance (foragers), maintained alive in ventilated plastic bags, and transported cold to the laboratory, where they were stored at −80 °C until processing [22]. In total, 108 samples (3 apiaries × 18 governorates × 2 seasons) were collected.

### 2.2. Varroa Mite Infestation Rate

We calculated the infestation rate by varroa mites of each sample. To do so, soap was mixed with warm water in a beaker with ca. 100 bees per sample retrieved from the −80 °C freezer(ThermoFisher Scientific, Frankfurt, Germany). The beaker was shaken and stirred for 10 min to improve the efficiency of dislodging varroa mites from the sample. The sample was then flushed with a large volume of warm water to separate the bees from the mites before pouring the fluid through a sieve (Impexron GmbH Pfullingen, Germany) with a mesh size of 3–4 mm to remove larger debris. To collect the mites, all of which should be washed through the first sieve, a second sieve (aperture 0.3 mm) was placed beneath the first. The mites that remained on the second sieve were counted, as were the washed bees in the sample, and the number of mites per 100 bees was calculated [23,24].

### 2.3. RNA Extraction and Detection of Virus 

Quantitative PCR (qPCR) was used to detect the presence or absence of ten common honey bee RNA virus targets: DWV-A DWV-B, BQCV, Chronic bee paralysis virus (CBPV), ABPV, SBV, SBPV, KBV, IAPV, and DWV-E (also known as Egyptian bee virus or EBV) using the primers listed in Appendix A and methods described in [25].

A selection of 10 frozen bees from each pooled sample of honey bee workers (150 bees from 5 hives) per apiary were taken for total RNA extraction, crushed in plastic RNase-free mesh extraction universal bags (BioReba, Reinach, Switzerland) with 500 µL of RLT-buffer per bee using an automated bee grinder, from which we recovered 200 µL of homogenate. This was mixed with 300 µL RLT-buffer containing 1% β-mercaptoethanol, then RNA was extracted from the homogenate using an RNeasy mini kit (Qiagen, Hilden, Germany) following the manufacturer’s instructions in a QiaCube robot (Qiagen, Hilden, Germany), as described in Tehel et al. [26]. The concentration of RNA was determined with an Epoch spectrophotometer (BioTek Instrument, Winooski, VT, USA). Then cDNA was synthesized using oligo-dT primers (Thermo Fisher Scientific, Schwerte, Germany) and reverse transcriptase (M-MLV and Revertase, Promega, Mannheim, Germany) following the manufacturer’s instructions. For cDNA synthesis, 800 ng of RNA were used, after which the resultant cDNA was diluted 1:10 prior to use in quantitative real-time PCR (qPCR). qPCRs were performed on a qPCR-QuantStudio3 (Thermo Fisher Scientific, Germany) using SYBR green Sensimix (Bioline, Luckenwalde, Germany) and the specific primers listed in Appendix A to determine virus presence or absence. We simultaneously amplified β-actin (AMActin/Actin related protein 1, primers described in Appendix A), an internal reference (‘housekeeping’) gene for honey bees that allowed us to independently ensure that RNA extraction and cDNA synthesis were successful and that the cDNA was amplifiable by qPCRs [27]; 95% (102 out of 108) of samples passed this quality control step (Cq < 30). qPCR amplification program steps were: 5 min at 95 °C, followed by 40 cycles of 10 s at 95 °C, 30 s at 57 °C (replaced by 54 °C for ABPV), and 30 s at 72 °C (including a read at each cycle), followed by 5 min at 72 °C. Thereafter, a melt curve prolife was run (one cycle of 95 °C for 1 min and 50 °C for 1 min followed by 50 °C to 95 °C at 0.5 °C per second increments) to check that the correct product had been amplified, i.e., an unambiguous, smooth melt curve profile peaking at the correct dissociation temperature. A Cq threshold of 35 was used to define a sample as positive (Cq < 35) to minimize the rate of false positives of the virus [25] and samples were run in duplicate. qPCRs were repeated for samples whose technical duplicate Cq values differed by >0.5 (ca. 5% of samples). Negative and positive controls were included in each qPCR plate. Positive controls for 9 of the 10 viral targets were isolated from qPCR-positive lab samples. There was no positive control for DWV-D available. The specificity and accuracy of qPCR products was validated for all samples by examining the melt curve at the end of the PCR program to ensure that only one product of the correct dissociation (melt) temperature was amplified in each reaction.

### 2.4. Statistical Analysis 

All statistical analyses and data visualizations were performed in GraphPad Prism 7.0 (GraphPad, La Jolla, San Diego, CA, USA). Normality of data was assessed by use of the Kolmogorov–Smirnov test and homogeneity of variance was determined with Levene’s test. The results were expressed as the number of varroa per 100 bees or the presence or absence of varroa mites, as well as viral presence/absence in each apiary per season. Chi-square tests were used to test for differences in proportions (varroa mite presence/absence; viral prevalence/absence in winter versus summer). The association between viral presence (yes/no) and varroa infestation (number of varroa per 100 bees) were compared using Student’s *t*-tests with Welch’s correction for unequal variances and Bonferroni correction for multiple testing (12 tests, representing the occurrence of a virus in the winter or the summer). An adjusted alpha level of 0.05 was used for all statistical tests to define significance. 

## 3. Results

### 3.1. Varroa Mite Presence per Sample and Infestation Rate

Varroa mites were found in 87% of colonies, a proportion which was identical in winter and summer seasons in 2021 (Chi-square test, n.s.; Figure 2). The highest rates of varroa mite infestation (mites per 100 bees) were 9% in bees collected from Faiyum governorate) and 14% for those collected from Minya governorate for winter and summer, respectively (Appendix A). There was no significant difference in the degree of varroa infestation (mites per 100 honey bees) between seasons (Student’s *t*-test, *p* > 0.05; Appendix A).

### 3.2. Prevalence of Honey Bee Viruses

DWV-A was the most prevalent virus among the ten honey bee viruses screened, followed by BQCV, ABPV, and CBPV. KBV and SBPV were detected only in the winter while IAPV and SBV were detected only in the summer (Figure 2). Surprisingly, DWV-B, though rapidly spreading worldwide [28], was not detected in samples collected either during winter or summer. DWV-D (Egypt bee virus), a fourth distinct major variant of DWV linked to honey bee mortality in Egypt in the 1970s [29], was also not detected (Figure 2). There were no significant differences (Chi-square test, *p* > 0.05) in the prevalence of DWV-A, BQCV, ABPV, and CBPV between seasons. 

Viral infection and co-occurrence in apiaries varied subtly between seasons (Supplementary Appendix A) and locations (Figure 3). For example, DWV-A was found in all governorates during the winter season, but not in Sharqia or the New Valley governorates during the summer, while BQCV was detected in all governorates during the summer season, but not in Behira or Luxor governorates during the winter (Figure 3). In addition, during the winter season, only DWV-A and SBPV were detected in apiaries in Behira governorate, while during the summer season, DWV-A, BQCV, ABPV, and SBV were detected in the same governorate. Furthermore, DWV-A, BQCV, ABPV, CPBV, and IAPV were detected in Sohag governorate in upper Egypt only during the summer season, while DWV-A and BQCV were only detected in north Sinai in the east of Egypt in summer (Figure 3).

### 3.3. Association between Viral Prevalence and Varroa Infestation

To determine whether increased varroa infestation was associated with the presence of a virus, we compared the number of varroa mites per 100 bees in colonies either with or without each virus. We found that apiaries that were infected with BQCV in the winter season held significantly more varroa mites than those in which we did not detect this virus (Student’s *t*-test, adjusted *p* < 0.05; Figure 4), indicating that there is a seasonal association between the intensity of infection of varroa and the presence of this virus.

## 4. Discussion

This is the first comprehensive study to determine the prevalence and distribution of ten widespread honey bee viruses in Egyptian honey bee colonies. Our findings revealed that DWV-A and BQCV were widely distributed in Egyptian honey bee colonies in both winter and summer, 2021. There was no difference in varroa infestation rates between seasons.

There are multiple genotypes of DWV, including the originally described and widely distributed genotype A [30], the rapidly spreading genotype B (otherwise known as *Varroa destructor* virus-1) [31,32], the more recently discovered but rarely detected genotype C [16,33], and genotype D (otherwise known as EBV), which is apparently no longer in circulation in honey bee populations [29]. Either DWV-A or DWV-B are the most common DWV variants worldwide due to their close relationship with and transmission by *V*. *destructor* [29]. In our study, DWV-A was the most frequently detected virus in all regions of Egypt during both winter and summer (80% and 74% of apiaries, respectively). Given that we sampled a pool of honey bees per apiary, DWV-A is likely present in most, if not all, colonies and apiaries of Egypt. That we did not detect DWV-D supports the result of de Miranda et al. (2022) [29] (lack of detection) for this genotype, first recorded in Egypt in the 1970s.

DWV was a rarely detected virus before the arrival of *V. destructor* into *A. mellifera* populations [34]. It remains at low prevalence in populations of *A. mellifera* devoid of *V. destructor* (e.g., [35]). The main mode of transmission of DWV between honey bees within a colony is nowadays horizontal, i.e., vector-based, by *V. destructor* [14,36,37,38]. One might therefore expect varroa infestation to be associated with the presence of DWV in a colony. However, we found colonies infected with DWV-A to have only a subtly and non-significantly higher varroa infestation (mites per 100 bees) in both winter and summer seasons compared to apiaries in which we did not detect this virus. 

Another intriguing finding was that DWV-B was not detected in any of our samples, despite the fact that DWV-A is seemingly being replaced by DWV-B, which is expanding its range worldwide [28,32] and already dominating in Europe [28,35,39] and increasing in prevalence elsewhere, (e.g., Israel [40] and USA [41]), potentially driving down DWV-A’s prevalence [28]. Despite the fact that DWV-B is better adapted than DWV-A for transmission by *V. destructor* [28], our interpretation for the lack of DWV-B in our study is that Egypt prohibits honey bee importation and that DWV-B may not yet have arrived in the country or, if it has, then only in a limited number of apiaries. It suggests that, if present in the country [21], DWV-B remains at very low prevalence and at a very low intensity of infection.

The second most prevalent virus among Egyptian honey bees was BQCV, which was detected in 63% (in winter) to 69% (in summer) of tested apiaries, followed by ABPV (37% and 44%, respectively), and CBPV (13% and 9%, respectively) in 2021. BQCV has also been reported as a highly prevalent virus in *Apis cerana* in Korea, where its occurrence in adult bees varied across geographic regions [42,43]. A high prevalence of BQCV infection of *A. mellifera* has also been detected in France [44], Japan [45], China [46], Germany [47], and the USA [48]. In contrast, a low prevalence of BQCV was reported in *A. mellifera* in Austria [49], Denmark [50], Jordan [51], and Brazil [52], as well as in Japanese *A. cerana* [45]. ABPV has also been found in several countries around the world and it is thought to be a common bee infective agent that is frequently detected in apparently healthy colonies [53]. It has, however, been proposed as the primary cause of bee mortality in colonies in Germany [54], Yugoslavia [55], France [56], and the United States of America [57] in the 1990s.

Why we found the presence of BQCV in a colony to be statistically associated with the level of varroa infestation in only the winter season is not clear because others have not found an association between them [58]. BQCV is not known to be transmitted by *V. destructor*; if it were to be vectored by varroa, it might speed up colony collapse by varroa mites vectoring BQCV to pupae and adults, where it can be highly virulent [59]. It is generally accepted in beekeeping practice that most diseases that are not directly linked to varroa mites may be statistically associated under adverse situations, such as when the colony is weak, after cold periods, and when sources of forage are scarce [60]. Therefore, it might be the case that the presence of BQCV in a colony is a marker for a diseased, stressed colony as a consequence of varroa parasitism, and in which BQCV is an opportunistic pathogen able to replicate in an immune-compromised colony.

The current study revealed multiple concurrent infections of honey bee viruses that varied subtly across location and season. Furthermore, some viruses found in bee colonies during the summer (e.g., IAPV and SBV) were not found in those colonies during the winter, and vice versa (e.g., KBV and SBPV). Our results are consistent with previous research that has also reported simultaneous multiple infections of honey bee viruses [46,61,62] and seasonal changes in bee viral infections in *A. mellifera* [46,63,64]. The latter could be attributed to climate conditions (winter vs. summer), which can have an immediate impact on a host’s lifestyle, or it can have an indirect impact on pathogen transmission [65]. Additionally, beekeeping practices such as hive placement, water provisioning, varroa mite control, and nutritional support [66] could significantly affect the health status and viral prevalence in colonies.

## 5. Conclusions

This is the first comprehensive study to determine the prevalence and distribution of ten common honey bee viral targets in Egyptian honey bee colonies. DWV-A and BQCV were widely distributed in Egyptian honey bee colonies in both winter and summer, 2021, whereas the DWV variant now spreading around the world, DWV-B, was not. Our data on current viral prevalence in honey bees could assist in the protection of Egypt’s beekeeping industry. Moreover, our study aids in the systematic assessment of the global honey bee virome by filling a knowledge gap about the prevalence of honey bee viruses in Egypt.

## Figures and Tables

**Figure 1 insects-14-00172-f001:**
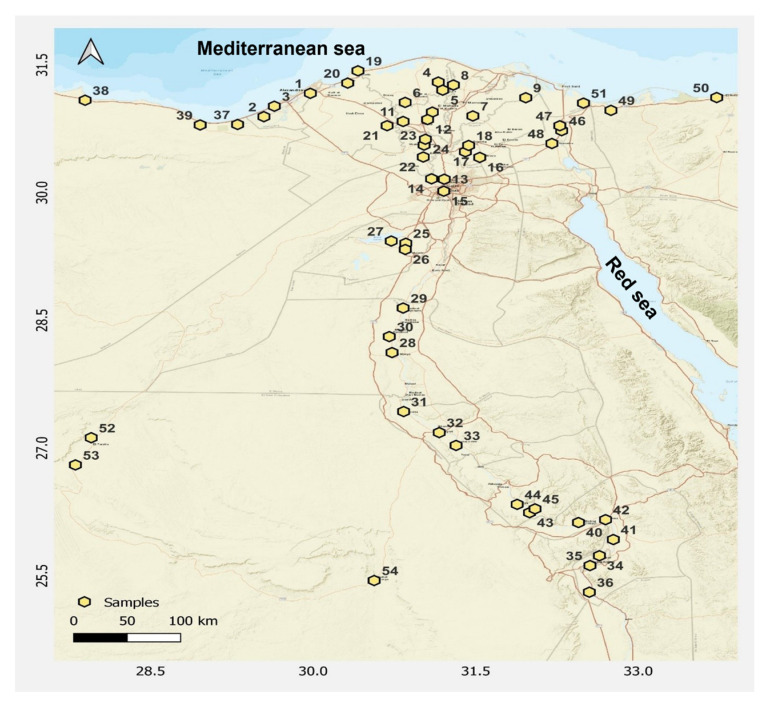
Distribution of the 54 apiaries in 18 governorates in Egypt, from each apiary of which a pool of honey bee workers was collected from 5 colonies; apiary numbers from 1 to 54 were from the following governorates: Alexandria (1, 2, 3), Kafr Elsheikh (4, 5, 6), Dakahlia (7, 8, 9), Gharbia (10, 11, 12), Cairo & Giza (13, 14, 15), Sharkia (16, 17, 18), El Beheira (19, 20, 21), Menofia (22, 23, 24), Faiyum (25, 26, 27), Minya (28, 29, 30), Asyut (31, 32, 33), Luxor (34, 35, 36), Matrouh (37, 38, 39), Qena (40, 41, 42), Sohag (43, 44, 45), Ismailia (46, 47,48), Sini (49, 50, 51), and New Valley (52, 53, 54) (produced by: UTM-GEO MAP-App).

**Figure 2 insects-14-00172-f002:**
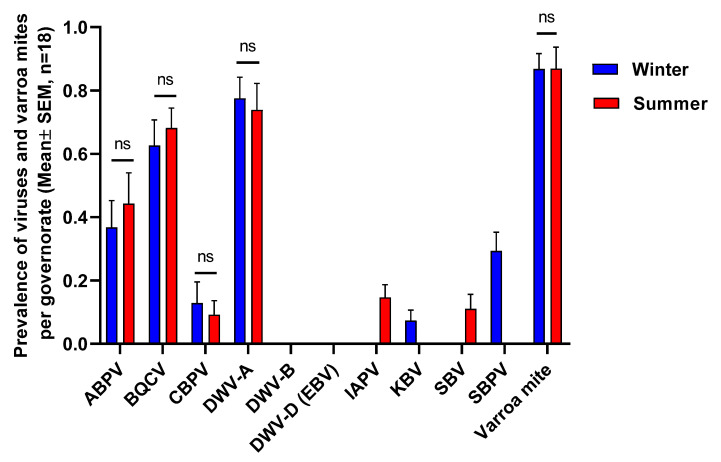
Prevalence of honey bee viruses (detected/not detected) and presence of varroa mites (detected/not detected) in honey bee samples collected from 54 apiaries across 18 governorates (3 apiaries per governorate, each apiary comprising 5 colonies) in Egypt during winter and summer, 2021. Columns represent the mean prevalence/presence per governorate (±SEM). There were no differences in virus prevalence or the probability of varroa being detected in a sample between winter and summer (Chi-square test, *p* > 0.05), ns denotes non-significant differences.

**Figure 3 insects-14-00172-f003:**
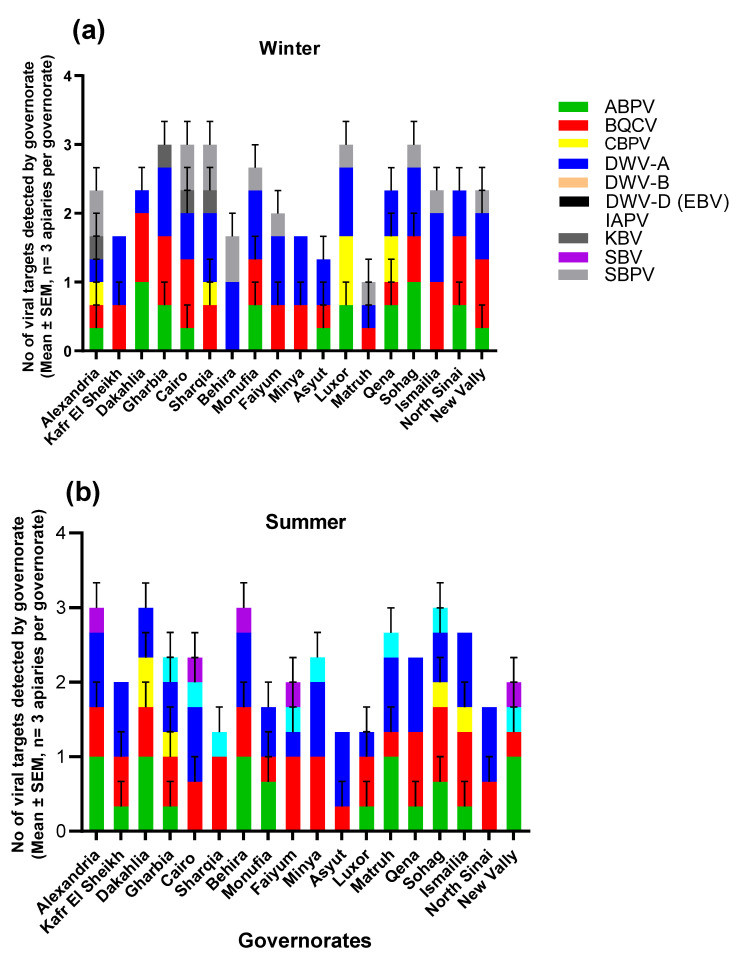
Number of viral targets detected by governorate (Mean ± SEM, n = 3 apiaries per governorate). During (**a**) winter and (**b**) summer, 2021, a total of 150 bees were collected from 5 colonies per apiary per location throughout 18 governorates in Egypt, of which 10 were analyzed.

**Figure 4 insects-14-00172-f004:**
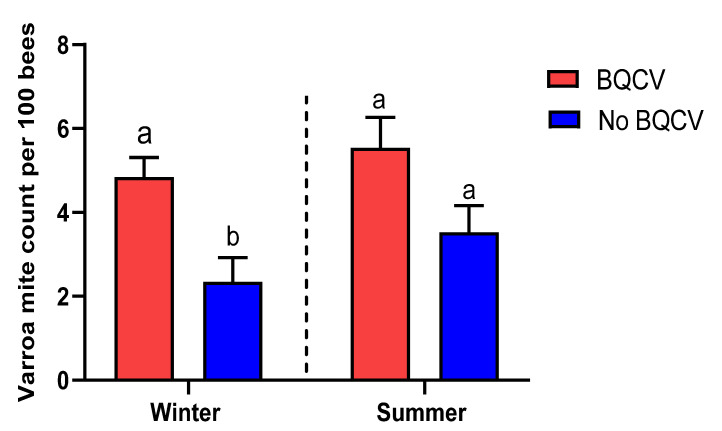
Association between viral prevalence and varroa infestation. Columns represent the mean count of varroa mites per 100 bees (± SEM). Different lower-case letters indicate a statistically significant difference (Student’s *t*-test with Bonferroni correction for multiple testing, adjusted *p* < 0.05) in varroa count between colonies infected or not infected with BQCV in the winter season.

## Data Availability

The data presented in this study are available in Appendix A.

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
