# Peer review of "Nationwide Screening for Bee Viruses in Apis mellifera Colonies in Egypt"

_insects, 2023, doi:10.3390/insects14020172_

Round 1

Reviewer 1 Report

The paper " Nationwide screening for bee viruses in Apis mellifera colonies in Egypt" describes the results of a survey of the prevalence of widespread bee viruses in Egyptian honey bee colonies, depending on geographical location, season or infestation with Varroa destructor (Varroa) mites. This study contributes to filling the knowledge gap on the prevalence of honey bee viruses in Egypt. The paper is interesting and well written.

First of all, I have no complaints about the introductory chapter, which provides sufficient information. Unfortunately, I have a few comments on the statistical methods and the discussion of the results.

In the 'Statistical analysis' section, the authors write that they used the Student's t-test to analyse the data, which is also mentioned in the 'Results' (line 182), but the Mann-Whitney test is mentioned in Figure S2. So, which test was used to compare viral infection in summer and winter?

The legend to Figure 3 states that 'Composite samples of honey bee workers (150 bees from 5 hives) were collected from each of three apiaries per location...'. However, in this virus study, instead of 150 bees, 10 bees from each apiary were used (line 134). Please correct this discrepancy.

Figure S1. Please specify what the boxplots and whiskers indicate.

Interestingly, the number of viruses detected in the same apiary differs between winter and summer (lines 184-190). I would like to see more discussion on why some viruses detected in summer in bee colonies were not detected in winter in those colonies and vice versa. What might have influenced the disappearance of the virus from the bee colony?

Author Response

Reviwer 1:

The paper " Nationwide screening for bee viruses in Apis mellifera colonies in Egypt" describes the results of a survey of the prevalence of widespread bee viruses in Egyptian honey bee colonies, depending on geographical location, season or infestation with Varroa destructor (Varroa) mites. This study contributes to filling the knowledge gap on the prevalence of honey bee viruses in Egypt. The paper is interesting and well written.

First of all, I have no complaints about the introductory chapter, which provides sufficient information. Unfortunately, I have a few comments on the statistical methods and the discussion of the results.

*******RESPONSE

We thank the referee for thier feedback and below we responded to their comments

**********

In the 'Statistical analysis' section, the authors write that they used the Student's t-test to analyse the data, which is also mentioned in the 'Results' (line 182), but the Mann-Whitney test is mentioned in Figure S2. So, which test was used to compare viral infection in summer and winter?

*******RESPONSE

We thank the referee for pointing this out and apologize for any confusion this has caused. The varroa infestation (number of varroa per 100 bees) in colonies with or without a virus‘ presence (yes/no) was compared using Student’s t-test. (Though the differences in viral Cqs in each apiary per season shown in former Fig.S2 were compared using a Mann Whitney U-test (because the data were not normally distributed), to address referee 2's concern about qPCR and virus detection, we decided to delete these data from supplementary materials. We therfore deleted and edited the text: Line 156-161, Line 189-192).

**********

The legend to Figure 3 states that 'Composite samples of honey bee workers (150 bees from 5 hives) were collected from each of three apiaries per location...'. However, in this virus study, instead of 150 bees, 10 bees from each apiary were used (line 134). Please correct this discrepancy.

*******RESPONSE

The referee is correct. During winter and summer 2021, a pool of 150 bees was collected from five colonies per apiary, of which 10 were analyzed. We revised the text and the legand of figure 3 accordingly.

**********

Figure S1. Please specify what the boxplots and whiskers indicate.

*******RESPONSE

We aplogize for this missing information. Each boxplot shows the median and interquartiles, the whiskers show 95% confidence intervals. We added this information to the legand of figure S1, as suggested.

**********

Interestingly, the number of viruses detected in the same apiary differs between winter and summer (lines 184-190). I would like to see more discussion on why some viruses detected in summer in bee colonies were not detected in winter in those colonies and vice versa. What might have influenced the disappearance of the virus from the bee colony?

*******RESPONSE

We thank the referee for their comment. Viruses were probably present in both summer and winter but were below detection threshold when ‘not detected’. However, in response to the referee‘s comment, we add further discussion to the variation in prevalence of viruses in the text (line 285-295).

**********

Reviewer 2 Report

Review Report: 

This study describes a screening effort of 10 honey bee viruses conducted on 105 samples of Apis mellifera sampled from Egypt (most probably A. m. lamarckii which is the native subspecies of this area). Varroa load (per 100 bees) was assessed in these samples as well. Subsequently, authors carried out basic locality and seasonal comparisons for both varroa and viruses.  

This work is composed of two main components: 1- quantification by qPCR of honey bee viral loads and 2- varroa mite count in 100 bees. The hardcore of this work relays on the qPCR technique, which unfortunately was poorly executed and displayed in this study. A qPCR should lead to proper and accurate quantifications of the target genes (virus in this case), which should be displayed in a form of either Relative quantity (RQ) or normalized expression of the (RQ) across the dataset. RQ is calculated by obtaining the ddCt values, RQ = 2(-ddCt). The reason ddct values are required here (not Cq normalized against actin only), is that an inter plate calibrator is needed as >2 plates were run. 

Major Concerns: 

1-    qPCR were not conducted using the gold standard method and no inter plate calibrator was given. Authors solely relayed on comparing Cq values / or dCq values against actin.  

2-    The use of a single housekeeping gene (Actin) is troubling. The recommendation is 2-3 stable housekeeping genes for acceptable accuracy and quantification of the targets. 

3-    In Figure 2: varroa load (count) and virus prevalence (RQ) cannot be displayed on the same y-axis since, the first value is a count and the second is a relative quantification (scale).    

4-    Figure 3 comprises no error bars to assess the stability of the dataset and is poorly displayed for qpcr data as stated above.  

5-    Causality cannot be established by mere comparisons of means. A proper correlation analysis should be conducted between viral and varroa loads, or conducting regression line with values of coefficient regression in order to assess such link.  

Additional comments to help authors in future submissions of their manuscript: 

L19: Why “so-called” stressors? There is enough evidence linking colony loss with various biotic and abiotic  stressors.

L69: delete “so-called”.  

Table1: qPCR efficiency exceeding 100% does not make sense. Revise this information. 

Table1: If these primers have already been published in ref 31, 32, 34, 35, why would they need to be detailed again in this paper.  

Authors called upon Student’s t-test for pairwise comparisons, although not mentioned in the text, it is not clear if the dataset followed a normal distribution or not (similar viral qPCR data usually don’t). Authors used Chi2 values, which lead to believe that they used Kruskal-Wallis non parametric test due to failure in normality apparently. Thus, for pairwise comparison, a non-parametric test should be used instead of a t-test.  

Author Response

Reviewer 2:

This study describes a screening effort of 10 honey bee viruses conducted on 105 samples of Apis mellifera sampled from Egypt (most probably A. m. lamarckii which is the native subspecies of this area). Varroa load (per 100 bees) was assessed in these samples as well. Subsequently, authors carried out basic locality and seasonal comparisons for both varroa and viruses.   

This work is composed of two main components: 1- quantification by qPCR of honey bee viral loads and 2- varroa mite count in 100 bees. The hardcore of this work relays on the qPCR technique, which unfortunately was poorly executed and displayed in this study. A qPCR should lead to proper and accurate quantifications of the target genes (virus in this case), which should be displayed in a form of either Relative quantity (RQ) or normalized expression of the (RQ) across the dataset. RQ is calculated by obtaining the ddCt values, RQ = 2(-ddCt). The reason ddct values are required here (not Cq normalized against actin only), is that an inter plate calibrator is needed as >2 plates were run.

 *******RESPONSE

We understand the concerns of the referee; indeed, when quantifying viral titres, we have formerly used a standard curve on each qPCR plate as a means of quantification, which we dd not do in this study of Egyptian honey bees. We therefore expunged any reference to quantification of viral load from the ms. Instead, we now revise the ms to state that we used qPCR (as a very sensitive and repeatable method) to detect the presence of a virus. We set our threshold of detection at Cq = 35; Cq values greater than 35 were considered negative, as is standard practice (de Miranda et al. 2013). The honey bee reference gene that we simultaneously qPCRed was used as a positive control that our RNA extraction, cDNA synthesis and qPCR had functioned properly.

De Miranda, J. R., Bailey, L., Ball, B. V., Blanchard, P., Budge, G. E., Chejanovsky, N., ... & Van Der Steen, J. J. (2013). Standard methods for virus research in Apis mellifera. Journal of apicultural research, 52(4), 1-56.

 *************

Major Concerns: 

1-    qPCR were not conducted using the gold standard method and no inter plate calibrator was given. Authors solely relayed on comparing Cq values / or dCq values against actin. 

*******RESPONSE

We agree with the referee that it would be more accurate to quantify viral load for positive samples for each viral target using standards (e.g. with ten 10-fold dilutions of the respective PCR product covering the observed concentrations) if we wanted to explore the intensity of infection of different viruses. But, in that case, we most likely would have analysed individual bees or bees pooled from just one colony so as to calculate viral titre per bee or per colony. This would have resulted in a much more expensive study which would have been superfluous to our aim of determining the distribution of viruses in Egyptian honey bees. Instead, as described in response to the comment above, we now explain in the ms that we performed a qualitative analysis (presence/absence of PCR product) to detect the prevalence of viruses following standard methods for virus research in Apis mellifera (de Miranda et al., 2013).  To avoid misunderstandings, we removed the data presented in Figure S2 from the supplementary materials and edited the text. (Line 156-163, Line 189-192).

************

2-    The use of a single housekeeping gene (Actin) is troubling. The recommendation is 2-3 stable housekeeping genes for acceptable accuracy and quantification of the targets.

 *******RESPONSE

We understand and accept the referee's point and now explain that we used amplification of beta-action merely as a ‘positive control’ of our methodology and integrity of the RNA of each sample. Beta-actin is a very reliable and dependable reference gene and sufficient to demonstrate that RNA has been extracted efficiently, cDNA has worked, and PCR/qPCR functions as expected.

************

3-    In Figure 2: varroa load (count) and virus prevalence (RQ) cannot be displayed on the same y-axis since, the first value is a count and the second is a relative quantification (scale).  

 *******RESPONSE

We apologise for the confusion we have caused by our poor wording. We now make clear that Fig. 2 gives the prevalence of viruses and also the presence of varroa mites per colony. We changed the x-axis legend to vary between 0 and 1. We also changed the title of section 3.1 to:

3.1. Varroa mite presence per sample and infestation rate

We also changed the legend to Fig. 2 to clarify what was plotted to make sure that readers are aware that we plot presence/absence per sample. The legend now reads:

Figure 2. Prevalence of honey bee viruses (detected/not detected) and presence of varroa mites (detected/not detected) in honey bee samples collected from 54 apiaries across 18 governorates (three apiaries per governorate, each apiary comprising 5 colonies) in Egypt during winter and summer 2021. Columns represent the mean prevalence/presence per governorate (± SEM). There were no differences in virus prevalence or the probability of varroa being detected in a sample between winter and summer (Chi-square test, P > 0.05).

******* 

4-    Figure 3 comprises no error bars to assess the stability of the dataset and is poorly displayed for qpcr data as stated above.  

*******RESPONSE

We thank the referee for pointing this out. Figure 3 has been modified to show number of viral targets detected by governorate (Mean ± SEM, n= 3 apiaries per governorate) as recommended.

**********

5-    Causality cannot be established by mere comparisons of means. A proper correlation analysis should be conducted between viral and varroa loads, or conducting regression line with values of coefficient regression in order to assess such link.  

*******RESPONSE

We understand the referee's point of view. However, because we did not quantify viral load per colony in the current study, we are unable to perform the suggested correlation analysis. Instead, we can only perform an association analysis, for which we use chi-square tests.

**********

Additional comments to help authors in future submissions of their manuscript: 

L19: Why “so-called” stressors? There is enough evidence linking colony loss with various biotic and abiotic stressors. edited

L69: delete “so-called”.  Done

Table1: qPCR efficiency exceeding 100% does not make sense. Revise this information. 

*******RESPONSE

Efficiency is recommended to lie from 85-115% for the primers and qPCR conditions to be acceptable (Svec et al., 2015).

Svec, D., Tichopad, A., Novosadova, V., Pfaffl, M. W., & Kubista, M. (2015). How good is a PCR efficiency estimate: Recommendations for precise and robust qPCR efficiency assessments. Biomolecular detection and quantification, 3, 9-16.

***********

Table1: If these primers have already been published in ref 31, 32, 34, 35, why would they need to be detailed again in this paper.  

*******RESPONSE

As suggested, we moved primer details to the supplementary materials (Table S1 in excel file). 

**********

Authors called upon Student’s t-test for pairwise comparisons, although not mentioned in the text, it is not clear if the dataset followed a normal distribution or not (similar viral qPCR data usually don’t). Authors used Chi2 values, which lead to believe that they used Kruskal-Wallis non parametric test due to failure in normality apparently. Thus, for pairwise comparison, a non-parametric test should be used instead of a t-test.  

*******RESPONSE

We thank the referee for their comment and apologize for the missing information. The association between viral presence (yes/no) and varroa infestation (number of varroa per 100 bees) were compared using Student’s t-tests and the data was normally distributed. However, we used 2 X 2 contingency Chi-square tests to test for differences in proportions (varroa mite presence/absence; viral presence/absence) in winter versus summer. We, therefore, edited section (2.4. Statistical analysis) to address the referee's comment.

**********

Reviewer 3 Report

The manuscript entitled “Nationwide screening for bee viruses in Apis mellifera colonies in Egypt” is a simple work, however very well written and comprehensive. I only found minor flaws and will recommend minor changes.

The authors should verify some formatting issues, for instance, Fig. 1 (line 86) was not in bold.

While Fig. 2 (line 161) is in bold.

Why "Fig" for the main figures and "fig" for supplementary figures?

Lines 31-32: Can the authors add P-values?

Lines 117-133: Is this a table 1 footnote or the main text?

The font size has changed. It makes more sense to include it in the main text.

Line 119: Why did the authors choose this gene? Can the authors add a reference?

Lines 160-161: Can the authors provide the number for winter and summer? or is it exactly the same?

It says that the difference is non-significant but is not clear if 87% is for summer winter or both.

Line 161: missing a space after 2021

Line 162: Where are these samples from? It should be written in the main text.

Line 163: Can we also see these numbers in sup tables 1 and 2?

Section 3.2: The authors should also describe the results considering the geographical distribution of the different viruses

Maybe it would also be interesting to describe where more viruses (different viruses) were found

Line 175: Can the authors tell the geographic distribution?

Section Discussion: Can the authors add a sentence discussing the results of ABPV?

Author Response

Reviewer 3:

The manuscript entitled “Nationwide screening for bee viruses in Apis mellifera colonies in Egypt” is a simple work, however very well written and comprehensive. I only found minor flaws and will recommend minor changes.

*******RESPONSE

We thank the referee for their positive evaluation of the MS.

**********

The authors should verify some formatting issues, for instance, Fig. 1 (line 86) was not in bold.

While Fig. 2 (line 161) is in bold. Done

Why "Fig" for the main figures and "fig" for supplementary figures? Done

Lines 31-32: Can the authors add P-values? Done

Lines 117-133: Is this a table 1 footnote or the main text? The font size has changed. It makes more sense to include it in the main text. Done

Line 119: Why did the authors choose this gene? Can the authors add a reference? Done

Lines 160-161: Can the authors provide the number for winter and summer? or is it exactly the same? It says that the difference is non-significant but is not clear if 87% is for summer winter or both.

*******RESPONSE

The answer is: for both. We now clarify that the % is the same in winter and summer. The text now reads:

Varroa mites were found in 87 % of colonies, a proportion which was identical in winter and summer seasons in 2021 (Chi-square test, n.s.; Fig. 2). (Line 167-168)

***********

Line 161: missing a space after 2021 Done

Line 162: Where are these samples from? It should be written in the main text. Done

Line 163: Can we also see these numbers in sup tables 1 and 2? We highlighted these numbers in yellow colours in (Tables S2&S3) in supplemental data files.

Section 3.2: The authors should also describe the results considering the geographical distribution of the different viruses. Maybe it would also be interesting to describe where more viruses (different viruses) were found.

*******RESPONSE

We thank the referee for their comment. As suggested, more information has been added and discussed (line 194-197; 285-295).

***********

Line 175: Can the authors tell the geographic distribution?

*******RESPONSE

We mentioned this information (line 170).

************

Section Discussion: Can the authors add a sentence discussing the results of ABPV?

*******RESPONSE

As suggested, we discussed the results of ABPV (line 268-273).

************

Reviewer 4 Report

It is good work and has interesting results. I suggest the authors explain more in detail about the materials and methods. 

Author Response

Reviewer 4:

It is good work and has interesting results. I suggest the authors explain more in detail about the materials and methods.

*******RESPONSE

Thanks for your positive evaluation on the MS.  Materials and methods have been revised, and more information has been added.

**********

Round 2

Reviewer 2 Report

L19: Delete “so-called”.

L68: Delete “so-called”.

L114: Table 1 does not exist anymore. 

L123: delete “Then”.

L132-133: Actin is not to “check” the success of your cDNA synthesis or RNA extraction quality, it is a housekeeping gene to standardize your cq values among samples and plates. Please correct accordingly. 

L142: “…positive controls…” where did this come from? Did you run a positive control for each of the ten viruses? Clarify this crucial point as of the origin of a validated viral positive control for your 10 viruses (plasmid copy, lab samples..etc). 

L145: “though we …..viral identity” delete this sentence.    

Author Response

Responses to reviewers

Comments and Suggestions for Authors

L19: Delete “so-called”. Done

L68: Delete “so-called”. Done

L114: Table 1 does not exist anymore. Edited to Table S1.

L123: delete “Then”. Done

L132-133: Actin is not to “check” the success of your cDNA synthesis or RNA extraction quality, it is a housekeeping gene to standardize your cq values among samples and plates. Please correct accordingly.

********Response

The referee is in principle correct that actin (housekeeping gene) is used for standardizing Cq values across plates. But as referee 2 of the first version of the ms pointed out very forcefully in their review, it is desirable to have more than one reference housekeeping gene to standardize Cq values and to quantify viral titres in a sample. In response to referee 2’s comments, we therefore removed the standardization of Cq values across plates and we removed all reference to the estimation of absolute or relative viral titres in the revised ms now under review. It is therefore inaccurate in the current version of the ms to state that we used the housekeeping gene to standardize Cq values.

What is accurate is that we used the Cq values of actin as a form of quality control to ensure that we had succeffuly extracted RNA, successfully synthesised cDNA and that the cDNA was amplifiable by qPCR. We have therefore removed the word “check” from the ms and added the following sentence (L133-136):

“We simultaneously amplified β-actin (AMActin /Actin related protein 1, primers described in Supplementary Table S1), an internal reference (‘housekeeping’) gene for honey bees that allowed us to independently ensure that RNA extraction and cDNA synthesis were successful and that the cDNA was amplifiable by qPCRs [27]; 95% (102 out of 108) of samples passed this quality control step (Cq<30)”.

************

L142: “…positive controls…” where did this come from? Did you run a positive control for each of the ten viruses? Clarify this crucial point as of the origin of a validated viral positive control for your 10 viruses (plasmid copy, lab samples.etc).

********Response

For 9 of the viruses, yes, we had a positive control (a sample that was positive by qPCR for the virus). We did not have a positive control for DWV-D, and it seems that the virus has disappeared from the world so we suspect that nobody has one, unless it is a synthetic molecule based on the putative sequence of DWV-D. The 9 positive controls were isolated from our current or former lab samples. As suggested, we included this information in the text (line 147 -149), which now reads:

“Positive controls for nine of the ten viral targets were isolated from qPCR-positive lab samples. There was no positive control for DWV-D available”.

************

L145: “though we …..viral identity” delete this sentence.   Done
